# Therapeutic Hypothermia and Its Role in Preserving Brain Volume in Term Neonates with Perinatal Asphyxia

**DOI:** 10.3390/jcm13237121

**Published:** 2024-11-25

**Authors:** Hernán Felipe García Arias, Gloria Liliana Porras-Hurtado, Jorge Mario Estrada-Álvarez, Natalia Cardona-Ramirez, Feliza Restrepo-Restrepo, Carolina Serrano, David Cárdenas-Peña, Álvaro Ángel Orozco-Gutiérrez

**Affiliations:** 1SISTEMIC Research Group, University of Antioquia, Medellín 050010, Colombia; 2Caja de Compensación Familiar de Risaralda, Salud Comfamiliar, Pereira 660003, Colombia; gporras@comfamiliar.com (G.L.P.-H.); jestradaa@comfamiliar.com (J.M.E.-Á.); ncardonar@comfamiliar.com (N.C.-R.); 3Hospital Pablo Tobón Uribe, Medellín 050010, Colombia; 4Clinica Universitaria, Universidad Pontificia Bolivariana, Medellín 050010, Colombia; carolina.serrano@upb.edu.co; 5Automatics Research Group, Technologic University of Pereira, Pereira 660003, Colombia; dcardenasp@utp.edu.co (D.C.-P.); aaog@utp.edu.co (Á.Á.O.-G.)

**Keywords:** perinatal asphyxia, therapeutic hypothermia, cerebral volumetry, neurodevelopmental outcomes

## Abstract

**Background:** Perinatal asphyxia is a major cause of neonatal morbidity and mortality, often resulting in hypoxic-ischemic encephalopathy (HIE) with long-term neurodevelopmental impairments. While therapeutic hypothermia has emerged as a promising intervention to reduce brain damage, its specific impact on key brain structures and long-term neurodevelopmental outcomes remains underexplored. This study aims to evaluate the effects of therapeutic hypothermia on brain volumetry, cortical thickness, and neurodevelopment in term neonates with perinatal asphyxia. **Methods:** This prospective cohort study enrolled 34 term neonates with perinatal asphyxia, with 12 receiving therapeutic hypothermia and 22 serving as controls without hypothermia. Brain MRI data were analyzed using Infant FreeSurfer to quantify the basal ganglia volumes, gray matter, white matter, cerebellum, cortical gyri, and cortical thickness. Neurodevelopmental outcomes were assessed at 18 and 24 months, using the Bayley Scale III, evaluating the motor, cognitive, and language domains. Genetic analyses, including next-generation sequencing (NGS) and microarray testing, were performed to investigate potential neurodevelopmental markers and confounding factors. **Results:** Neonates treated with hypothermia demonstrated significantly larger gray and white matter volumes, with a 3.7-fold increase in gray matter (*p* = 0.025) and a 2.2-fold increase in white matter (*p* = 0.025). Hippocampal volume increased 3.4-fold (*p* = 0.032) in the hypothermia group. However, no significant volumetric differences were observed in the cerebellum, thalamus, or other subcortical regions. Moderate correlations were found between white matter volume and cognitive outcomes, but these associations were not statistically significant. **Conclusions:** Therapeutic hypothermia appears to have region-specific neuroprotective effects, particularly in gray and white matter and the hippocampus, which may contribute to improved neurodevelopmental outcomes. However, the impact was not uniform across all brain structures. Further research is needed, to investigate the long-term benefits and to optimize therapeutic strategies by integrating advanced neuroimaging techniques and genetic insights.

## 1. Introduction

Hypoxic-ischemic encephalopathy (HIE) is a significant cause of neonatal morbidity and mortality globally, with an incidence of 1 to 8 cases per 1000 live births, rising to over 10 per 1000 live births in low- and middle-income countries (LMICs), due to limited access to perinatal and neonatal care [1,2,3]. This high incidence substantially burdens healthcare systems and affected families, particularly in resource-limited settings. The extent and the nature of brain injuries caused by HIE depend on the severity and duration of ischemic episodes, with acute cases impacting critical brain areas, such as the deep perirolandic nuclei and hippocampal regions, and prolonged or partial ischemia affecting cerebral intervascular basins [4,5,6]. Moderate or intermittent hypoxia often results in mixed injury patterns, affecting nuclei and the cortex along with intervascular regions [7,8]. These variations underscore the complexity of HIE and highlight the need for timely, targeted therapeutic interventions [9].

Therapeutic hypothermia has become the standard intervention for neonates with HIE, reducing brain damage by lowering metabolic rates and inhibiting neuronal apoptosis [10,11]. Hypothermia is thought to provide neuroprotection by mitigating the effects of reactive oxygen species (ROS) and pro-inflammatory mediators, critical contributors to HIE pathophysiology [5,12,13]. In neonates, this treatment involves reducing the body temperature to 33.5 ± 0.5 °C, typically applied through whole-body cooling, which has become the standard method, as selective head cooling is no longer widely used. Studies have shown no significant differences in efficacy between whole-body and head cooling, and selective cooling devices have primarily been discontinued. Whole-body cooling offers a uniform approach and is generally applied in specialized neonatal intensive care units (NICUs) [14].

While therapeutic hypothermia remains the most widely accepted treatment, other pharmacologic and experimental interventions for HIE are currently under investigation. For example, studies are exploring the neuroprotective potential of xanthines, xenon gas, and stem cell therapies, with initial findings indicating promising effects on reducing neural damage. Erythropoietin (EPO), once considered a promising neuroprotective agent, is no longer recommended, based on recent evidence [15]. These alternative treatments, while not yet achieving the same level of clinical acceptance as hypothermia, provide a broader context for therapeutic strategies in HIE and may inform future approaches, particularly as adjuncts to hypothermia or in settings where hypothermia is not feasible [15].

Despite the established benefits of therapeutic hypothermia, access remains limited in LMICs, including countries in Latin America, leading to disparities in clinical outcomes [8]. Studies conducted in these settings highlight the logistical and infrastructural challenges that prevent the widespread implementation of hypothermia therapy, which may affect the generalizability of results obtained from studies in high-income countries. However, the HELIX trial offers essential insights into therapeutic hypothermia (TH) application in low- and middle-income countries (LMICs). Conducted in an LMIC setting, the HELIX study examined the efficacy of TH, ultimately finding no significant benefit. For instance, Serrano-Tabares et al. [16] explored outcomes in Colombian patients with HIE who received hypothermia, noting regional differences in organic behavior and the challenges of consistent therapy, due to limited resources [16]. Additionally, Gutierrez et al. [17] investigated the correlation between ammonium levels and HIE severity, suggesting that hypothermia’s effects may vary under LMIC conditions, further illustrating the complexity of translating high-income therapeutic protocols to resource-limited settings [17]. These findings underscore the need for tailored approaches to therapeutic hypothermia in LMICs, where access and implementation barriers may impact the effectiveness of this intervention and limit the generalizability of standard protocols.

Most studies on therapeutic hypothermia have focused on primary outcomes, such as mortality and disability. However, there remains a significant gap in knowledge regarding the specific impact of hypothermia on brain volumetric changes and neurodevelopmental outcomes. Conventional MRI findings often appear normal in some neonates treated with hypothermia. Nevertheless, these patients may still experience adverse outcomes, including behavioral disorders and learning difficulties that hypothermia therapy does not seem to mitigate. This raises the question of whether subtle volumetric differences, undetectable on standard imaging, could help explain these outcomes. Identifying such volumetric markers could provide insights into why specific neurodevelopmental sequelae persist despite hypothermia and help refine therapeutic strategies [18,19]. Addressing this knowledge gap is critical to refining therapeutic approaches for neonates with HIE and understanding the broader neuroprotective potential of hypothermia.

Advances in neuroimaging, particularly magnetic resonance imaging (MRI), have enabled detailed assessments of brain structure following hypoxic events [20,21]. Tools like Infant FreeSurfer allow for automated, precise quantification of brain morphometric measures specifically adapted to the unique challenges of infant imaging, such as smaller regions of interest, motion artifacts, and varying contrast due to ongoing myelination and maturation processes. These tools address the gap in infant imaging capabilities by offering robust segmentation and surface extraction designed for infants aged 0–2 years, enabling accurate comparisons of cortical features like volume, thickness, surface area, and curvature [22]. Such capabilities are particularly valuable as atypical cortical morphometry has been linked to neuropsychiatric and developmental disorders, highlighting the need for advanced imaging tools in neonatal brain studies. Combined with machine learning algorithms that enhance segmentation accuracy and volumetric assessments, these methods provide a comprehensive picture of brain recovery, making them particularly suitable for studying neonatal brain development after HIE [8].

This study aimed to quantitatively evaluate the effects of therapeutic hypothermia on brain structure volumes in neonates with perinatal asphyxia and to explore its implications for neurodevelopmental outcomes. We hypothesize that therapeutic hypothermia leads to region-specific neuroprotective effects, resulting in significant volumetric differences in gray matter, white matter, and hippocampal regions, which may correlate with improved neurodevelopmental outcomes. By assessing quantitative volumetric changes in these key brain structures and examining their associations with neurodevelopmental outcomes, this study sought to provide new insights into hypothermia’s potential neuroprotective benefits. Advanced MRI analyses with Infant FreeSurfer, machine learning techniques, and genetic evaluations were used to compare treated neonates to a control group without hypothermia treatment, focusing on quantitative structural and developmental outcomes. These findings aim to contribute to optimizing treatment protocols for neonates with HIE and improving their long-term quality of life.

## 2. Materials and Methods

A prospective cohort study involving term neonates with perinatal asphyxia. Two groups were formed: the exposed group consisted of patients from a specialized healthcare center (HPT) that adhered to clinical practice guidelines for therapeutic hypothermia in neonates with perinatal asphyxia, with the protocol detailed in Section 2.2. The unexposed group consisted of patients from another healthcare facility (CF) where clinical guidelines for therapeutic hypothermia were not implemented and, therefore, they received standard clinical care at the attending physician’s discretion for each individual case. MRI was performed at a time point close to the perinatal insult and again at two years post-event. Clinical data such as complications, birth characteristics, and maternal history were collected and used as control variables in the analysis.

### 2.1. Participants

Prospective cohort study of 34 asphyxiated newborns, with 12 receiving hypothermia treatment and 22 not receiving it. These newborns were admitted to two neonatal intensive care units in the central region of Colombia from January 2015 to December 2023. The two healthcare centers involved in this study followed standardized national criteria for diagnosing hypoxic-ischemic encephalopathy (HIE). This consistency in case selection aimed to ensure uniformity across the study cohort, minimizing the potential for center-specific variations. Due to this uniform adherence to HIE diagnostic criteria, data points such as Sarnat scores and pH values were not individually collected. The newborns were selected according to the criteria for HIE indicated by the American College of Obstetricians and Gynecologists, such as (a) umbilical cord arterial pH less than 7, (b) Apgar score between 0 and 5 for longer than 5 min; (c) neurological manifestations, such as seizures, coma, or hypotonia; and (d) multisystem organ dysfunction (e.g., cardiovascular, gastrointestinal, hematological, pulmonary, or renal system). In addition, the severity of its manifestation/neurological damage was evaluated according to the modified SARNAT scale (i.e., clinical staging of HIE), MRI assessment, and Bayley Scale III.

### 2.2. Therapeutic Hypothermia Procedure in Newborns with HIE

Therapeutic hypothermia is a standard treatment for full-term newborns diagnosed with hypoxic-ischemic encephalopathy (HIE), to reduce the risk of brain injury. This procedure involves lowering the baby’s core body temperature to help decrease the metabolic demands of the brain cells and to limit the extent of neuronal damage caused by lack of oxygen [11]. Therapeutic hypothermia can be administered through whole-body cooling or selective head cooling. During whole-body cooling, the baby’s temperature is lowered to 33.5 ± 0.5 °C, using specialized cooling blankets or devices. With selective head cooling, the temperature is maintained at 34.5 ± 0.5 °C, using a cooling cap placed over the baby’s head, which allows for targeted temperature control while keeping the rest of the body closer to normal temperature levels [23].

The cooling process typically starts within six hours of birth, to maximize the neuroprotective effects. The target temperature is maintained for 72 h, followed by a gradual rewarming phase, usually at 0.5 °C per hour, until the baby’s temperature reaches 36.5 °C. Close monitoring of vital signs, including heart rate, respiratory function, and electrolyte levels, is essential throughout the procedure, to manage potential side effects, such as bradycardia and electrolyte imbalances. Therapeutic hypothermia aims to slow down the cascade of biochemical processes that lead to cell death, including reducing apoptosis and inflammatory responses. It ultimately offers a neuroprotective effect in newborns affected by HIE. While it has shown efficacy in improving survival rates and reducing long-term neurological impairments, its successful application depends on precise temperature control and careful patient monitoring by trained healthcare providers [24].

### 2.3. Neurodevelopmental Assessment

Neurodevelopmental outcomes were assessed longitudinally, using the Bayley Scales of Infant and Toddler Development, Third Edition (Bayley-III), a standardized tool widely used to evaluate the developmental functioning of infants and toddlers [25]. The Bayley-III assesses three core domains: cognitive (i.e., problem-solving abilities and understanding of basic concepts), motor (i.e., fine and gross motor skills), and language skills (i.e., receptive and expressive language abilities).

Evaluations were conducted at 18 and 24 months, representing the minimum follow-up intervals recommended for a comprehensive Bayley assessment. These time points are ideal for capturing key neurodevelopmental milestones, allowing for a reliable and standardized evaluation of cognitive, motor, and language outcomes. Trained neuropsychologists conducted all the assessments, to ensure the consistency and reliability of the results. The scores from each domain were analyzed to determine any significant differences between the neonates who received hypothermia therapy and those who did not, with adjustments made for potential confounding variables such as gestational age and severity of HIE.

### 2.4. Genetic Analysis

To determine whether the clinical phenotype of each patient could be associated with genetic alterations potentially acting as a masker for hypoxic-ischemic encephalopathy (HIE), we conducted next-generation sequencing (NGS) on a panel of genes related to neurotransmitters for 34 patients. Additionally, microarray comparative genomic hybridization (CGH) was performed on ten patients exhibiting phenotypic alterations or neurological compromise, to identify DNA copy number imbalances, including deletions and duplications, and to conduct whole exome sequencing. Blood samples were collected from all patients between 2017 and 2024 at Comfamiliar Risaralda and processed by GENCEL PHARMA COL in Bogotá. The DNA amplification and bioinformatic analyses were carried out by an external laboratory, ensuring the objectivity and impartiality of the study. The authors were not involved in the sequencing or variant identification processes. Variants identified through these analyses were classified into two categories, providing a clear understanding of the findings: (i) pathogenic variants (PVs), and (ii) variants of unknown significance (VUSs).

### 2.5. Brain Volume Analysis

The study used magnetic resonance imaging (MRI) with a 1.5T Siemens Heltenier system to assess brain volume. In Colombia, access to higher-resolution 3T MRI scanners is limited, as only a few hospital centers are equipped with this technology. Consequently, the 1.5T scanner was selected based on accessibility and standard imaging practices within our region. High-resolution T1-weighted images were obtained using a 3D magnetization-prepared rapid gradient-echo (MPRAGE) protocol. The acquisition parameters were as follows: repetition time = 2400 ms, echo time = 3.5 ms, inversion time = 1000 ms, flip angle = 10°, field of view = 256 × 256 mm, acquisition matrix = 320 × 320, 192 slices, and resolution = 1.0 × 1.0 × 1.0 mm. MRI images were collected between 2 and 3 years of age, to assess brain structure development post-treatment. All clinical team and imaging analysts were blinded to group allocation, to reduce potential bias in clinical evaluations and MRI analyses.

Preprocessing of the images included quality control, motion correction, and spatial normalization to a Montreal Neurological Institute (MNI) template. Cortical parcellation was carried out using the “infant FreeSurfer” framework, explicitly designed for processing infant brain images. This framework facilitates precise segmentation and measurement of cortical and subcortical structures, addressing specific challenges in infant brain imaging, such as smaller regions of interest and variable contrast properties due to ongoing myelination [22].

A senior neuroradiologist labeled all the MRI volumes regarding the brain injury, in terms of location and severity. Data analysis was focused on total brain volume, total white matter volume (WMV), gray matter volume (GMV), and specific sub-cortical regions of interest derived from the parcellation process such as basal ganglia and hippocampus. This detailed approach allowed us to gain a comprehensive understanding of the brain’s structure and function.

### 2.6. Statistical Analysis

An exploratory and descriptive data analysis was conducted, to examine the distribution of the calculated brain volumes. Given the non-normal distribution and positively skewed nature of the brain volume variables, this analysis identified a generalized linear model (GLM) with a gamma link function as the most appropriate approach for detecting differences between groups. Confounders were selected based on an extensive literature review identifying variables known to influence brain volume outcomes. Relevant factors, such as genetic mutations, were included due to their documented associations with brain development and volumetric variations. This selection aimed to control for potential confounding effects, enhancing the model’s capacity to accurately assess the relationship between hypothermia exposure and brain volumetric outcomes.

The model adjusted for confounders, including the presence or absence of genetic mutations, age at the time of MRI, birth characteristics, and delivery complications. The results are presented as regression coefficients, effect sizes, corresponding confidence intervals, and *p*-values. Welch’s *t*-tests, which do not assume equal variances, were also applied for group comparisons. Additionally, a Pearson correlation matrix was generated to assess the strength and direction of these relationships. All statistical analyses were performed using R software (version 4.2), with a significance level set at 0.05.

## 3. Results

### 3.1. Descriptive Results

A total of 34 newborns were included in the study. The sex distribution was similar between both groups, with 42% females in the hypothermia group and 50% in the non-hypothermia group (p=0.6). The rest of the demographics can be seen in Table 1.

Table 2 presents the maternal and prenatal characteristics of the study cohort. The use of epidural anesthesia appeared more frequent among mothers in the hypothermia group, with 33% reporting its use compared to 14% in the non-hypothermia group (p=0.2). Similarly, 17% of mothers in the hypothermia group reported using medication during pregnancy, compared to 4.5% in the non-hypothermia group (p=0.3). While these differences did not reach statistical significance, they may reflect variations in clinical practices and patient populations between the two centers involved in the study. Thus, maternal and prenatal characteristics were relatively balanced between the groups, suggesting that these factors are unlikely to have significantly influenced the outcomes of hypothermia treatment. Finally, for a detailed summary of maternal and newborn features, including perinatal factors, readers can refer to Appendix A, Table A1, which provides comprehensive data on the distribution.

### 3.2. Hypothermia Effect

Table 3 presents the results from the adjusted regression models, which assessed the volumetric changes in different brain structures in newborns subjected to hypothermia compared to those without hypothermia. The models adjusted for confounding factors, such as genetic mutations, age at the time of MRI, birth characteristics, and delivery complications.

Gray matter showed a coefficient of 1.314, indicating that newborns exposed to hypothermia exhibited an average gray matter volume approximately 3.7 times larger than those without hypothermia (*p* = 0.025). Similarly, white matter volume had a coefficient of 0.798, suggesting that white matter was 2.2 times larger in the hypothermia group (*p* = 0.025). The WM/GM ratio showed a 3.0-fold increase (*p* = 0.026). These findings show bigger brain volumes for these structures in the hypothermia group. Additionally, the hippocampus exhibited an increase in volume (3.4-fold, *p* = 0.032).

However, other regions, such as the cerebellum, caudate nucleus, putamen, thalamus, globus pallidus, amygdala, and nucleus accumbens, did not show statistically significant differences between the groups (*p* > 0.05). While some structures demonstrated modest volume increases, these were not statistically robust, indicating a less clear relationship between hypothermia and these brain regions.

These findings indicate that hypothermia may have a potential effect on brain volume in certain regions but does not lead to clinically or statistically significant differences in neurodevelopmental outcomes as measured by the Bayley Scale.

Figure 1 shows the correlations between neurodevelopmental outcomes (cognitive, language, and motor functions) and brain volume (total brain, gray matter, and white matter) in two groups: children treated with hypothermia and those who were not. In the hypothermia group, moderate positive correlations were noted between cognitive function and both white matter volume (r = 0.57) and total brain volume (r = 0.46), though these associations were not statistically significant. This suggests a potential trend where larger brain volumes, particularly in white matter, may support better cognitive outcomes in children who receive hypothermia therapy. Additionally, a strong correlation was observed between motor and language functions (r = 0.96), indicating that those neurodevelopmental domains may have been closely linked in this group. However, weak correlations between brain volumes and language function (r = −0.03 to 0.17) suggest that brain structure alone may not have been a key determinant of language outcomes in these children. In the non-hypothermia group, correlations between neurodevelopmental outcomes and brain volumes were weak. Cognitive functioned showed minimal correlation with total brain volume (r = 0.11) and white matter volume (r = 0.06), and similar patterns were observed for language and motor functions. This lack of significant associations suggests that brain volume may have less influence on neurodevelopmental outcomes in the absence of hypothermia. These findings highlight the potential protective effect of hypothermia on brain structures like white matter, which may contribute to better cognitive and motor outcomes. However, the absence of statistically significant results across both groups calls for further research with larger sample sizes to better understand these relationships.

### 3.3. Cortical Thickness and Structural Differences Between Hypothermia and Non-Hypothermia Groups

Figure 2 compares brain volumes in neonates with and without therapeutic hypothermia, using density plots of gray matter, white matter, hippocampus, and the white matter/gray matter (WM/GM) ratio, along with cortical and subcortical renderings. The plots show that the neonates treated with hypothermia (blue) tended to have higher volumes of gray and white matter than the untreated neonates (pink), suggesting potential preservation of these tissues. The WM/GM ratio was also higher in the hypothermia group, indicating better overall brain preservation. Notably, the hippocampus, important for memory and cognitive functions, showed a broader range of larger volumes in the hypothermia group, supporting the idea that hypothermia may offer neuroprotective effects. Visualizations of cortical surfaces further emphasize the structural differences, indicating that hypothermia helps maintain cortical integrity. Overall, these results suggest that therapeutic hypothermia may help preserve key brain structures after perinatal asphyxia. This preservation could potentially lead to improved neurodevelopmental outcomes, reinforcing the potential benefits of this treatment. However, further analysis is needed to confirm the significance and clinical impact of these findings.

The cortical thickness for the hypothermia-treated group and the non-hypothermia group was compared, using 3D surface reconstructions (Figure 3). In the hypothermia-treated group (Figure 3 right), a heatmap of cortical thickness revealed a higher degree of preservation, with thicker regions primarily located in the central sulci and occipital areas. The cortical thickness ranged from 0 mm to 5 mm, with the highest values represented in red and the lowest in blue. These results suggest that therapeutic hypothermia promotes cortical growth or maintenance in these critical areas.

Conversely, the non-hypothermia group (Figure 3, left) exhibited more widespread cortical thinning, particularly in the frontal and parietal lobes. These regions, responsible for cognitive and motor functions, showed reduced cortical thickness, indicating the potential for developmental delays or impairments in this untreated population. The comparison of these two groups highlights the differential effects of therapeutic hypothermia, suggesting a neuroprotective effect in preserving cortical structure post-injury.

Figure 3 shows cortical thickness measurements in millimeters for two groups of term neonates: those who did not receive therapeutic hypothermia (left panel) and those who did (right panel). The scale ranges from purple (indicating thinner cortical areas) to red (indicating thicker cortical areas), with a scale of 0 to 5 mm. Visual comparison between the two groups suggests differences in the distribution and extent of cortical thickness across various brain regions. In the hypothermia-treated group, thicker regions (represented by green-to-red hues) appear more pronounced and widespread, particularly in motor and sensory processing areas. In contrast, the non-hypothermia group displays a greater prevalence of thinner regions (depicted in shades of blue and purple), suggesting that therapeutic hypothermia may help preserve or promote greater cortical thickness. This preservation of cortical structure might correlate with improved neurodevelopmental outcomes, as larger cortical thickness has been associated with better cognitive and motor functions in previous studies. However, further quantitative analysis would be required to confirm these observations and assess their statistical significance. Additionally, these visual insights highlight the importance of therapeutic interventions in mitigating the effects of hypoxic-ischemic encephalopathy (HIE) on brain development.

## 4. Discussion

The study included 34 newborns. Among the prenatal characteristics, pre-eclampsia was observed in 14% of the mothers in the non-hypothermia group. At the same time, no cases were reported in the hypothermia group (*p* = 0.5), suggesting no significant association between pre-eclampsia and the application of hypothermia therapy. Although individual data on Sarnat scores and pH levels were not collected, both centers adhered to standardized national criteria for the diagnosis of HIE. This uniform approach minimized potential biases related to center-specific variations in case definition and selection.

Volumetric assessments of the brain, particularly in neonatal populations, provide crucial insights into the potential long-term neurological outcomes following perinatal events such as hypoxic-ischemic encephalopathy (HIE). Our study’s findings on the effects of therapeutic hypothermia on brain volumes in neonates with perinatal asphyxia are of significant importance. Brain volumes, measured through techniques like MRI, allow for the quantification of gray and white matter, which is integral to understanding the effects of HIE and therapeutic interventions like hypothermia. Clinically, larger brain volumes have been associated with better neurodevelopmental outcomes, while reductions in volume often correlate with cognitive and motor impairments [4].

It is well established that HIE can significantly reduce brain volumes in untreated infants, particularly affecting critical structures like the basal ganglia and hippocampus [20]. Studies have consistently demonstrated that infants with HIE who do not receive therapeutic intervention show marked reductions in the gray and white matter volumes, with adverse neurodevelopmental consequences [26]. In comparing our findings with previous research, studies have shown varying results regarding the preservation of unaffected brain regions in neonates with HIE who undergo therapeutic hypothermia. For instance, Rivero-Arias et al. [27] observed that therapeutic hypothermia had long-term neuroprotective effects, particularly in preserving brain volume and supporting neurodevelopmental outcomes, aligning with our findings of increased gray and white matter volumes in treated neonates. Similarly, Shankaran [2] demonstrated that children who received hypothermia therapy showed better neurodevelopmental outcomes, with specific improvements in cognitive functions related to white matter integrity, which our study also highlights through the observed white matter volume preservation. These comparisons underscore the region-specific neuroprotective effects of therapeutic hypothermia and suggest that certain brain areas may inherently have higher resistance to hypoxic damage or benefit variably from the intervention [24].

The lack of significant volumetric changes in some brain regions could be attributed to their differential vulnerability to hypoxic damage and varying responses to therapeutic hypothermia. Certain regions, such as the cerebellum and caudate nucleus, may inherently have a higher resistance to hypoxic conditions, possibly due to their specific metabolic profiles or blood supply [8]. Additionally, these regions may not benefit as much from the neuroprotective mechanisms induced by therapeutic hypothermia, such as the reduction of apoptosis and inflammation [5]. This selective protection could explain why structures like the hippocampus, which is highly susceptible to ischemic damage, show more pronounced preservation with hypothermia, whereas other regions remain unaffected.

Moreover, the more robust response observed in white matter volume compared to other structures may reflect the higher metabolic activity of white matter during early brain development. White matter is particularly vulnerable to oxidative stress and excitotoxicity during hypoxic events, leading to more severe damage in untreated cases of HIE [4]. The observed increase in white matter volume in hypothermia-treated infants suggests that therapeutic hypothermia may play a role in reducing white matter injury by mitigating the effects of reactive oxygen species (ROS) and promoting myelin repair processes. These findings align with previous studies that emphasize the critical role of white matter integrity in supporting neurodevelopmental outcomes and highlight the importance of targeted neuroprotective strategies for preserving white matter during therapeutic interventions [20].

Our study did not find statistically significant correlations between brain volumes and cognitive, language, or motor outcomes. However, moderate correlations were observed between cognitive function and white matter volume (r=0.585) and total brain volume (r=0.425). The lack of statistical significance may be attributed to the relatively small sample size and incomplete neurodevelopmental assessments for some participants early in the study [7]. These findings suggest a need for larger cohorts and more extended follow-up periods to validate the potential association between brain volumes and neurodevelopment.

## 5. Conclusions

Our study contributes to the growing body of evidence supporting the neuroprotective effects of therapeutic hypothermia in neonates with perinatal asphyxia. Specifically, hypothermia-treated infants demonstrated increased volumes in key brain structures such as gray matter, white matter, and the hippocampus, which may translate into improved neurodevelopmental outcomes. However, the region-specific nature of these volumetric changes suggests that certain brain areas may be inherently more resilient to hypoxic damage or variably responsive to hypothermia, underscoring the need for further research to elucidate the mechanisms behind these differential effects.

These findings also highlight the value of advanced neuroimaging techniques, such as cortical parcellation, to identify subtle structural changes that may not be visible in conventional imaging. This approach holds promise for early identification of atypical volumetric patterns that could impact neurodevelopment, informing early interventions and contributing to refined clinical guidelines in neonatal care.

The study’s findings emphasize the need to address the practical limitations and resource constraints in implementing hypothermia treatment, especially in low- and middle-income countries, where logistical challenges in transferring patients to specialized centers may impact the effectiveness of this intervention. Additionally, while moderate associations between white matter volumes and neurodevelopmental outcomes suggest potential pathways for further investigation, longitudinal studies will be essential to assess the sustained effects of hypothermia on cognitive and memory functions, particularly about increased hippocampal volumes observed in treated neonates.

Finally, future research directions should explore the potential role of genetic markers related to inflammation, neuromuscular function, and metabolism, as these may influence the variability in neurodevelopmental outcomes among neonates treated with hypothermia. Investigating these markers alongside volumetric patterns across different pathologies could provide deeper insights into prognostic factors and optimize individualized treatment strategies, ultimately enhancing the quality of life for children affected by HIE.

## Figures and Tables

**Figure 1 jcm-13-07121-f001:**
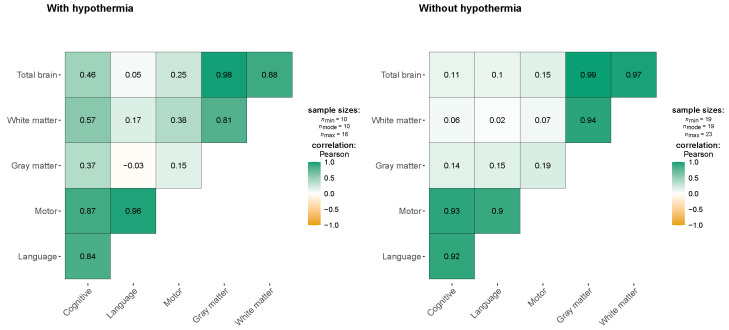
Correlation between neurodevelopment and brain volume.

**Figure 2 jcm-13-07121-f002:**
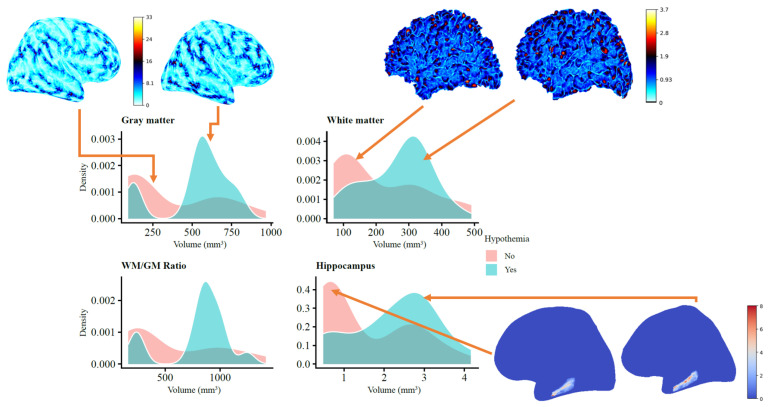
Brain volumetric comparison between neonates with and without therapeutic hypothermia, including density plots for gray matter, white matter, hippocampus volumes, and the WM/GM ratio, along with cortical renderings. The hypothermia-treated group (blue) shows a shift towards higher volumes in gray and white matter compared to the non-treated group (pink), suggesting tissue preservation.

**Figure 3 jcm-13-07121-f003:**
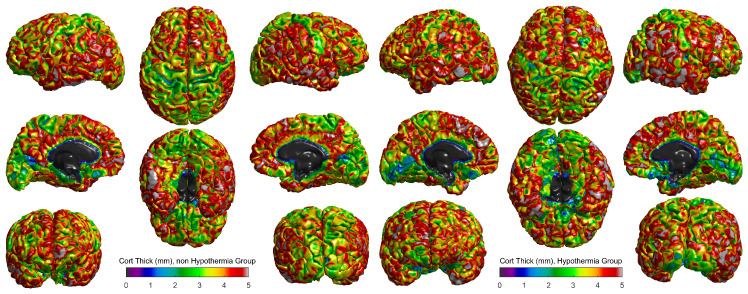
Average of cortical thickness (mm) for both groups.

**Table 1 jcm-13-07121-t001:** Population data for the hypotermia study.

Hypothermia				
**Characteristic**	**Yes, N = 12 ^1^**	**No, N = 22 ^1^**	**Overall, N = 34 ^1^**	***p*-Value ^2^**
Sex				
Female	5(42%)	11(50%)	16(47%)	0.6
Male	7(58%)	11(50%)	18(53%)	
Gestational age (weeks)	39.20 (38.8–39.2)	39.35 (38.0–40.0)	39.20 (38.2–40.00)	0.8
Mother’s age (years)	26.0 (24.0–30.2)	29.5 (24.5–33.8)	28.0 (24.0–33.0)	0.3
Prenatal control	11 (92%)	22 (100%)	33 (97%)	0.4

^1^ n (%); Median (IQR); ^2^ Pearson’s Chi-squared test; Wilcoxon rank sum test; Fisher’s exact test.

**Table 2 jcm-13-07121-t002:** Maternal and prenatal factors.

Characteristic	Yes, N = 12	No, N = 22	Overall, N = 34	*p*-Value
Pre-eclampsia				0.5
No	12 (100%)	19 (86%)	31 (91%)	
Yes	0 (0%)	3 (14%)	3 (8.8%)	
Chronic hypertension				
No	12 (100%)	22 (100%)	34 (100%)	
Smoking during pregnancy				>0.9
No	12 (100%)	21 (95%)	33 (97%)	
Yes	0 (0%)	1 (4.5%)	1 (2.9%)	
Previous cesarean section				
No	12 (100%)	22 (100%)	34 (100%)	
Epidural anesthesia				0.2
No	8 (67%)	19 (86%)	27 (79%)	
Yes	4 (33%)	3 (14%)	7 (21%)	
Intra-uterine growth retardation				
No	12 (100%)	22 (100%)	34 (100%)	
TORCH				
No	12 (100%)	22 (100%)	34 (100%)	
Medication during pregnancy				0.3
No	10 (83%)	21 (95%)	31 (91%)	
Yes	2 (17%)	1 (4.5%)	3 (8.8%)	
Maternal infections				>0.9
No	12 (100%)	21 (95%)	33 (97%)	
Yes	0 (0%)	1 (4.5%)	1 (2.9%)	
Gestational diabetes				
No	12 (100%)	22 (100%)	34 (100%)	
Primiparity				0.5
No	7 (58%)	10 (45%)	17 (50%)	
Yes	5 (42%)	12 (55%)	17 (50%)	
Fentanyl use				0.4
No	11 (92%)	22 (100%)	33 (97%)	
Yes	1 (8.3%)	0 (0%)	1 (2.9%)	
Other				0.7
No	7 (58%)	15 (68%)	22 (65%)	
Yes	5 (42%)	7 (32%)	12 (35%)	

**Table 3 jcm-13-07121-t003:** Adjusted regression models for brain structure volumes.

Brain Structure	Coefficient	Std. Error	Effect (Coefficient)	*t* Value	*p* Value	AIC
Gray matter	1.314	0.530	3.7	2.478	**0.025**	395.508
White matter	0.798	0.321	2.2	2.483	**0.025**	341.883
WM/GM ratio	1.086	0.443	3.0	2.452	**0.026**	412.594
Left cerebellum	0.037	0.050	1.0	0.749	0.464	161.359
Right cerebellum	0.077	0.082	1.1	0.937	0.363	188.329
Total cerebellum	0.061	0.075	1.1	0.819	0.425	224.077
Caudate	0.526	0.331	1.7	1.587	0.132	104.479
Putamen	0.146	0.078	1.2	1.868	0.080	40.092
Thalamus	0.119	0.064	1.1	1.870	0.080	50.563
Globus pallidus	0.332	0.195	1.4	1.702	0.108	15.003
Hippocampus	1.218	0.519	3.4	2.347	**0.032**	90.461
Amygdala	0.624	0.310	1.9	2.012	0.061	0.863
Accumbens	0.384	0.214	1.5	1.792	0.092	−39.749

Bold are to highligh the indicator variables among study.

## Data Availability

The data that support the findings of this study are available from the corresponding author upon reasonable request.

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
