# Peer review of "Therapeutic Hypothermia and Its Role in Preserving Brain Volume in Term Neonates with Perinatal Asphyxia"

_jcm, 2024, doi:10.3390/jcm13237121_

Round 1
Reviewer 1 Report
Comments and Suggestions for Authors
This manuscript presents a well-structured and thorough investigation into the neuroprotective effects of therapeutic hypothermia in neonates with perinatal asphyxia. The study provides valuable insights into the volumetric changes in key brain structures, such as gray matter, white matter, and the hippocampus, and examines their potential links to neurodevelopmental outcomes. The methodology is detailed, with appropriate statistical analyses and advanced neuroimaging techniques. The tables and figures are informative, effectively illustrating the findings. The discussion is well-grounded in existing literature, offering thoughtful interpretations of the results.
Overall, this study makes a meaningful contribution to the field. With the following refinements, its impact and clarity for readers will be further enhanced.
o The introduction is well-structured and effectively introduces the clinical significance of perinatal hypoxic-ischemic encephalopathy (HIE) and the role of therapeutic hypothermia. However, the flow of information could be streamlined for improved readability. Consider condensing some sentences to eliminate redundancy and enhance the clarity of core messages.
o Please include a brief summary of the burden of HIE globally (perhaps with specific incidence rates) would help readers unfamiliar with the field grasp the scope of the condition.
o The discussion on hypothermia’s neuroprotective mechanisms is valuable. However, it might be beneficial to briefly outline any other commonly used or experimental interventions for HIE, in addition to therapeutic hypothermia, to provide broader context.
o It would strengthen the argument to emphasize the gap in knowledge regarding volumetric and neurodevelopmental effects in neonates treated with hypothermia, providing a clear rationale for why this study is necessary.
o Although advancements in neuroimaging and genetic analysis are mentioned, I recommend briefly explaining why these tools (e.g., Infant-FreeSurfer and machine learning algorithms) are particularly suitable for studying neonatal brain development.
o The description of hypothermia temperature ranges could benefit from clarification. It may be helpful to specify the settings in which whole-body versus selective head cooling is generally applied and how these choices impact treatment outcomes.
o The brief mention of disparities in access to therapeutic hypothermia in low- and middle-income countries is important, though it could be elaborated upon. Discussing how limited access may impact the generalizability of study results could provide a more comprehensive understanding of the issue.
o The objectives of the study are clearly stated; however, I recommend refining them by specifying the study’s primary hypothesis. This addition would provide readers with a clearer framework for understanding the research focus and anticipated outcomes.
o The paragraph outlining the study’s aim would benefit from added specificity. I recommend explicitly stating that the study will evaluate "quantitative" differences in key brain regions. This clarification will help distinguish this research from prior studies focused on more qualitative neurodevelopmental outcomes and enhance the reader’s understanding of the study's unique contribution.
o Line 54, In the phrase "Therapeutic hypothermia in neotanes involves reducing- the word "neotanes" should be corrected to "neonates
o Line 55, There is inconsistency in the notation of temperature ranges “34.5 ± 0.5C”
o When mentioning “the protocol detailed later in the manuscript,” it would be helpful to either specify the exact subsection or briefly outline the protocol.
o The statistical analysis in the "Materials and Methods" section is comprehensive, and the use of a generalized linear model (GLM) with a gamma link function is appropriate for handling the non-normal distribution of brain volume data. However, it would be helpful to clarify the rationale for selecting specific confounders, such as genetic mutations, and to provide additional detail on how these confounders were identified. This clarification would strengthen the understanding of the model's design and enhance the rigor of the statistical approach.
o Please include a justification for selecting a 1.5T scanner rather than higher-resolution systems, as this choice impacts image quality and segmentation accuracy.
o Clarifying the rationale for choosing 18 and 24 months as assessment time points would add value, as neurodevelopment can be assessed at various stages.
o It might be useful to clarify the units for “Gestational age” and “Mother age” in Table 1
o Line 217-225, The text mentions that epidural anesthesia and medication during pregnancy were more frequent in the hypothermia group, though not statistically significant. Including a brief sentence discussing potential reasons for these differences (even if non-significant) could add depth to the interpretation.
o The discussion acknowledges the absence of significant volumetric differences in regions like the cerebellum and caudate nucleus. This observation is valuable, but expanding on the possible reasons, that would provide a more comprehensive understanding of why these regions may not benefit as much from hypothermia.
o In discussion, please include a brief comparison with previous studies that reported similar or differing results regarding these unaffected brain regions that could provide context and add to the interpretation.
o In conclusion, add a closing statement on the potential impact of this research on clinical guidelines for neonatal care, particularly in terms of early intervention, would give readers a better sense of the study’s translational potential.
o Consider including potential limitations of hypothermia treatment and practical challenges for implementing it in resource-limited settings.
o Highlight potential future research directions with specific imaging techniques and the role of genetic markers to address response variability.
Reviewer 2 Report
Comments and Suggestions for Authors
I have read this paper on brain volumetry in neonates either or not undergoing therapeutic hypothermia with great interest, and with a background on clinical perinatal research, including in this subpopulation. I do have concerns on the current analysis and report as provided, as the authors are - respectfully - rather overselling their findings.
In the introduction, I do miss some information on the HELIX trial (LMIC setting, TH was negative), with a NNT is likely appropriate to be mentioned. I agree that the majority of studies has focussed on clinical relevant outcome variables, although there is quite some supportive information on MRI or EEG as prediction tools for outcome (eg Weeke score, J Pediatr 2018, among others). In this way, your paper rather adds to the existing literature, and further support other data.
In my reading, you have to further stress the limitations of the study design, as it is well known that the 'center' effect, as well as 'circumstantial' handling and the subsequent outcome is relevant for this disease. I do miss data on (dis)similarities in eg neonatal 'picture', like Sarnat, or Thompson score, or pH/lactate. It is not clear if these 'obvious' clinical covariates were included in the adjusted regression model(s).
How similar are both cohorts ? Where there any a priority assumption, primary outcome and/or power calculation, or shoulld I understand your analysis as explorative ?
Finally, I would suggest to rephrase the ethics statement, as this is not really a case report, and consent is not limited to reporting, but also covers data acquisition and analysis (was MRI part of routine clinical care, or study related)
When (at what age) were MRI images collected (i have missed this crucial information), and were assessors blinded for group allocation (as the study was prospective). This is relevant for both the clinical assessment, as well as the MRI imaging.
minor, typo on line 54.
Round 2
Reviewer 2 Report
Comments and Suggestions for Authors
the comments have been well addressed, nothing to add